# Beliefs and Violent Behavior in Interpersonal Relationships of Young Adults: A Systematic Review

**DOI:** 10.3390/ijerph21111500

**Published:** 2024-11-12

**Authors:** Eduardo Araújo, Anita Santos, Claúdia Oliveira, Olga Souza Cruz, Diana Moreira

**Affiliations:** 1Department of Social Sciences and Behavior, University of Maia—UMAIA, Avenida Carlos Oliveira Campos, Castêlo da Maia, 4475-690 Maia, Portugal; anitasantos@umaia.pt (A.S.); claudiaoliveira.psic@gmail.com (C.O.); d011379@umaia.pt (O.S.C.); dianapatmoreira@gmail.com (D.M.); 2Center for Psychology, University of Porto, 4099-002 Porto, Portugal; 3JusGov—Justice and Governance Research Centre, University of Maia (UMAIA), 4710-057 Braga, Portugal; 4Faculty of Human Sciences of the Portuguese, University of Catholic, 1649-023 Lisboa, Portugal

**Keywords:** beliefs, attitudes, violent behavior, relationships, young adults

## Abstract

Beliefs are information-processing structures formed along an individual’s developmental pathway. Beliefs can legitimize involvement in inappropriate or violent behaviors, particularly when they crystallize into cognitive schemas. While beliefs aid individuals in interpreting the surrounding world, overly rigid and inflexible beliefs can constrain the individual’s ability to process available information. This Systematic Review, carried out according to the PRISMA norms and guidelines, aims to understand the most prevalent beliefs regarding relationships among young adults and to examine their associations with violent or deviant behaviors. Articles included in this review were retrieved from the EBSCO, PubMed, and Web of Science databases in July 2022, resulting in a total of 594 studies, which were subsequently screened by two independent reviewers. A total of 51 studies were then selected for full reading, but 36 were excluded based on pre-defined eligibility criteria, leaving a final sample of 18 studies published between 2014 and 2022. The main objectives, country of origin, instruments used, sample composition and age, main results and conclusions were extracted from each study. Findings point toward the presence of related and legitimate beliefs about violence in intimate relationships, domestic violence, sexual violence, acceptance of the rape myth, or consent to engage in sexual activities.

## 1. Introduction

From a very early age, people begin to develop relationships through their social interactions [1]. Firstly, with a limited number of people (e.g., parents). However, over the course of development, these relationships extend to others (e.g., friends) [2]. During these processes, individuals make observations, either consciously or not, which enable them to attain new knowledge [1]. In turn, this is internalized and stored in cognitive structures, commonly called beliefs, which serve for attribute meaning and are formed during the individual’s developmental pathway [3]. These structures comprise concepts and abstractions about the self, others, or the world. Once formed, they allow individuals to interpret their environment, make inferences, and predict what might happen “here and now” or in similar future circumstances. As normative structures, beliefs help individuals to adopt the most acceptable behaviors in a given situation [4].

According to Young [5], beliefs may tend to become rigid and organized into cognitive schemas. A cognitive schema is an often rigid and inflexible information processing structure that can negatively affect the individual’s interactions. However, for adaptive functioning, the individual’s beliefs and cognitive schemas should be flexible, thus enabling appropriate interpretations that encompass all information relating to the events faced. Yet, on some occasions, this is not observed and, in such situations, due to erroneous readings, individuals may respond prematurely. This may be explained by evolutionary, ontogenetic, and phylogenetic factors in Mankind [6]. Throughout Mankind’s evolutionary process, the lack of resources was a constant. Thus, to maintain homeostatic balance, in an idiosyncratic way, individuals learned to manage the available resources and acquired the ability to function using the minimum required energy. This would have resulted in the ability to make decisions quickly, even before a small number of contextual cues, like a facial expression. This skill, necessary for survival in the early days of Mankind, was refined over time and may explain certain behaviors [7]. As a result, people can now make near-instant readings of the events they encounter. However, due to the required brevity for these readings, individuals may devalue or ignore important clues, and act prematurely [8]. Additionally, any unsuitable responses may be exacerbated by maladaptive or non-normative beliefs [9].

### 1.1. Maladaptive Beliefs

Maladaptive beliefs are dysfunctional patterns, usually related to atypical developmental pathways (e.g., victimization), neglect and/or unmet basic needs (e.g., affection) [5], damage to attachments processes [10], or cognitive distortions [3]. These beliefs are dysfunctional and potentially disabling concepts that may interfere with an individual’s life, leading to a negative view of oneself and others [11]. Maladaptive beliefs may be associated with psychopathology, personality disorders, or problematic relationships marked by disrespect, violence or aggressive behaviors [4].

### 1.2. Interpersonal Relationships

Human beings are innately social; thus, for their well-being, they need to establish different bonds, generally called relationships. The first relationships are focused on meeting basic needs (e.g., food, shelter), and family relationships are critical here [10]. According to Berger [2], belonging to a peer group is important for well-being, as it helps to define identity or personal preferences. Thus, during childhood, the first friendship networks are created. As individuals mature, new needs arise, prompting other types of relationships. For instance, in early adolescence, individuals may develop interest in sexuality and establish intimate relationships, which coexist with existing relationships [12]. By late adolescence, it is expected that individuals will have formed a support network comprising: (i) family-based relationships (e.g., the family, nuclear and extended); (ii) friendships; (iii) intimate relationships (e.g., with a romantic partner); and (iv) broader interpersonal relationships [4]. It should also be noted that these relationships aim to meet several and different needs [2]. However, conflicts may arise, especially where the developmental path has fostered maladaptive beliefs, making it challenging to define boundaries in relationships (e.g., it may be difficult to distinguish the limits of a relationship of friendship or intimacy). This could be the case during the transition into adulthood, a period for which successful coping is crucial for individuals [13], which can be especially challenging when maladaptive beliefs are present [4].

### 1.3. Young Adults

The age range that defines young adulthood—the target population of this study—is not consensual. For most authors, young adulthood begins at 18 and ends at 29 years e.g., [14]. However, other authors extend this range to 31 e.g., [15], 35 e.g., [16], or even 40 years [17]. Nonetheless, it is widely accepted that the transition from adolescence to adulthood is marked by numerous physical, physiological, and neurological changes associated with growth completion and neurological maturation [18].

### 1.4. Young Adults and Associated Issues

Understanding the challenges that young adults face is important, as many arise directly from such changes [13]. This transitional phase is further marked by adolescent typical behaviors (e.g., substance abuse, oppositional behaviors) [19]. Additionally, societies often impose various demands (e.g., academic success), and choices (e.g., career paths), for which individuals may feel unprepared [20].

Young adults are also expected to gain independence from their parents or to enter more “serious” intimate relationships and potentially constitute their own family. In normative cases, despite certain difficulties, most young people successfully overcome this developmental stage and establish their own identity [21]. However, not all young adults experience this success, since many faces significant obstacles to achieving their goals (e.g., lack of employment opportunities) [22]. Consequently, the number of challenges can increase competition among young people (e.g., in choosing partners), and may lead to conflicts, which can sometimes escalate into violent or deviant behaviors. In these situations, these individuals’ (mal)adaptive beliefs play a determining role in avoiding or engaging in unwanted behaviors (e.g., violence) [14].

### 1.5. Violent Behavior

Numerous definitions and types of violence are described in the literature e.g., [14]. Generally, however, violent behavior is defined as any intentional behavior that by action (e.g., assault), purpose (e.g., threat), or inaction (e.g., negligence), is intended to cause harm or exercise coercive control over others [23]. This concept is dynamic and context-dependent (i.e., it varies according to the historical time, the group, or the society analyzed). As a social construct, violence’s legitimacy depends on interpretations by those involved. Violence can be, for instance, physical or psychological [24], face-to-face or remote (e.g., using technology), or verbal (e.g., insults). However, due to its complexity, it is challenging to find a comprehensive definition, as violence is a very heterogeneous phenomenon, and any definition must account for factors, such as, the nature of the act and the relationship between the involved parties [25].

Despite multiple overlapping definitions, it is crucial to make some distinctions: (i) intimate partner violence, which involves a pre-existing intimate relationship [25]; (ii) domestic violence, which involves the mistreatment of cohabitants, including people or animals [26]; (iii) dating violence, common among teenagers in dating relationships [27]; (iv) sexual violence, which includes acts of a sexual nature attempted or completed, without the victim’s prior consent. In particular, sexual violence can be perpetrated by a stranger, friend, family member, or partner (e.g., boyfriend or husband) [28]. It should also be noted that sexual violence, particularly against women, can be reinforced by a set of generally false beliefs that aim to deny that assault has occurred (e.g., rape myths).

In summary, violence is an increasingly visible phenomenon that raises considerable social concern. The number of reported cases suggests decreasing societal tolerance. As such, this is a problem that requires more attention [29]. Meanwhile, beliefs are a well-studied construct (e.g., cognitive distortions, irrational beliefs) [3]. However, the topic of violence-legitimizing beliefs, particularly among young adults, has not received adequate attention in the scientific community. In order to prevent these violence-legitimizing beliefs, it is important to better understand how young people’s beliefs influence violent behavior. These considerations, alongside the high social impact of violence, justify the social and scientific relevance of this systematic literature review (SLR).

## 2. Method

### 2.1. Research Questions and Target Population

This SLR aims to answer the following research questions: (i) Are young adults’ interpersonal relationships constrained by maladaptive beliefs (MB)? and (ii) What types of MB, that promote conflicts or violence, are most prevalent in this population?

The target population for this SLR is young adults (i.e., 18 to 35 years old), as suggested by Araújo et al. [21].

### 2.2. Inclusion and Exclusion Criteria

The following inclusion criteria were defined: (i) empirical studies using quantitative methodology; (ii) peer-reviewed studies; (iii) studies addressing beliefs in interpersonal relationships; and (iv) studies targeting young adults. The following exclusion criteria were defined: (i) systematic literature reviews, with or without meta-analysis; (ii) theses and dissertations; (iii) book chapters and/or other theoretical studies; (iv) studies with participants experiencing psychopathology; (v) studies on topics outside beliefs in interpersonal relationships; and (vi) longitudinal and/or experimental studies.

### 2.3. Procedures

This SLR was conducted following the guidelines of the Preferred Reporting Items for Systematic Reviews and Meta-Analyses (PRISMA) [30]. The predefined search expression was applied in July 2022, focusing on the titles and abstracts of peer-reviewed articles in the following scientific databases: (i) EBSCO; (ii) PubMed; and (iii) Web of Science. No restrictions (e.g., linguistic, temporal, or geographical) were applied. Special attention was given to the quality of the selected articles, following the guidelines from Child Care and Early Education Research Connections (CCEERC), [31]. Table 1 provides the quality analysis of the included studies.

### 2.4. Search Expression

The following search expression was used, with adaptations to meet each database’s requirements: TI (belief* OR perception* OR view* OR attitude*) AND AB (violence OR aggression OR hostility OR violent OR anger OR “aggressive behavior”) AND AB (young adult*). No filters or constraints (e.g., linguistic or temporal) were applied. For EBSCO, a total of 120 studies (i.e., after removal of duplicates), published between 1956 and 2022, were identified. In PubMed, 323 articles were identified, also published between 1956 and 2022. In Web of Science, 145 articles (i.e., after the removal of duplicates) were identified, published between 1956 and 2022. A manual search yielded an additional six studies. This resulted in a total of 594 abstracts (including the six retrieved from the manual search), which were analyzed by two independent judges with a master’s degree in forensic and clinical psychology. All disagreements were resolved through consensus. The agreement degree between judges was assessed through *Kohen’ Kappa* (*K*). The observed value (*K* = 91.7%) was considered almost perfect [44]. Thus, 51 articles were selected for full-text review. Subsequently, 36 were found not to meet the predefined criteria, and were excluded. Consequently, 18 studies were included in this SLR (cf. Figure 1), published between 2014 and 2022, from which the following information was extracted: (i) author, year, and country of publication; (ii) aims; (iii) sample characteristics (e.g., number of participants); (iv) instruments used; and (v) main results and conclusions.

## 3. Results

Eighteen descriptive empirical studies were included, mainly from the United States of America (USA) (*n* = 7), followed by Spain (*n* = 3), Australia (*n* = 2), Germany (*n* = 1), South Africa (*n* = 1), Lithuania (*n* = 1), Israel (*n* = 1), United Kingdom (*n* = 1), and Turkey (*n* = 1). The samples, mostly constituted by women (*n* = 5956; 63.5%), ranged from *N* = 148 (Aušraitė and Žardeckaitė-Matulaitienė [33]) to *N* = 2395 (Bonilla-Algovia and Rivas-Rivero [35]), totalizing 9385 participants. Notably, the study by Sanchez-Prada et al. [42] included 200 individuals of both genders, of which only 50 could be included in the study, so all reported data referred solely to this subgroup.

The studies addressed attitudes, beliefs, and behaviors relating to various aspects of violence and consent: (i) intimate partner violence (*n* = 6; Dardis et al. [37]; Malka et al. [40]; McKool et al. [41]; Moss et al. [17]; Sanchez-Prada et al. [42]); (ii) dynamics underlying consent to engage in sexual activities (*n* = 3; Jozkowski et al. [15]; King et al. [39]; Shafer et al. [28]; (iii) watching pornographic content (*n* = 2; Bernstein et al. [34]; Krahé et al. [16]); and (iv) domestic violence (*n* = 2; Adıbelli et al. [32]; Rodriguez et al. [26]).

The remaining studies (*n* = 5) focused on different issues, such as: (i) irrational beliefs and conflict resolution (Aušraitė and Žardeckaitė-Matulaitienė [33]; (ii) warning sign behaviors (Towler et al. [43]); (iii) beliefs about aggression (Wagener and Padmanabhanunni [9]; (iv) sexual violence (Canan et al. [36]), and sexual cyber violence (Durán and Rodríguez-Domínguez [38]). Table 2 provides a summary description of all these studies, including their aims, country of origin, participant age ranges, instruments, main results and conclusions.

### 3.1. Intimate Partner Violence (IPV)

Six studies examined IPV-related attitudes, beliefs, and behaviors. The first, from the USA, conducted by Dardis et al. [37], sought to understand what types of behaviors are perceived as being violent by 703 university students. The results suggest that women report more perpetration of physical and psychological violence, while men tend to report significantly more sexual violence perpetration (*p* < 0.001). A significant effect of gender (*p* < 0.001) was also observed on the perception of IPV. Men tend to consider all forms of violence against women as less abusive. It was also observed that IPV perpetrated by men against women is considered more serious than the one perpetrated by women against men (*p* < 0.001). It was also observed that, as opposed to women, men without a history of victimization tended to consider IPV as more abusive (*p* < 0.05) than those without an identical history.

A second study, also from the USA, performed by McKool et al. [41] aimed to assess the relationship between peer perception of IPV and self-reported perpetration by 202 men. Results revealed that in the preceding 12 months, 36% and 67% of men reported having committed physical and sexual violence, respectively. In addition, 92% of the participants underestimated the seriousness of the acts committed (e.g., physical aggression). Others (38%) revealed and assumed the seriousness of the acts they perpetrated (e.g., strangulation). Among those who confessed to having practiced sexual violence, 27% reported having done so more than 20 times. The most reported type of violence was related to coercive acts (e.g., forcing sex without a condom). However, no sociodemographic variable was associated with IPV (*p* > 0.05), and most acts of violence were perpetrated by individuals under the age of 25. Regarding the relationship between the peers’ and the perpetrators’ perception, no significant connection was found (*p* > 0.05). Only two participants revealed that a colleague had committed a violent act. It was also found that having friends who supported sexual violence was associated (*p* < 0.05) with its perpetration. Regarding discordance, the results indicated that more than 33% of the participants underestimated their colleague’s involvement in violence. Additionally, some participants underestimated the involvement of colleagues in sexual violence, even based on their accounts. A positive association was also observed between poor peer perception of violent behavior and physical, sexual, and self-reported violence (*p* < 0.01). Among those who admitted to committing violence, 98% were not aware of peer involvement in identical acts. Thus, only about 2% of perpetrators adequately identified their peers’ involvement in violent behavior based on their reports. In short, it seems that those who misidentified or underestimated peer involvement would be more likely to report IPV perpetration, compared to those who correctly identified peer misbehavior.

Also from the USA, a study conducted by Moss et al. [17] sought to understand the cognitions, social stereotypes, and gender beliefs associated with the IPV acceptance of 369 individuals. Results suggest a strong acceptance of women’s sexual objectification, followed by beliefs associated with traditional gender roles. On the opposite side, less acceptance of violence and stereotypes about black women was observed. The gender analyses suggest that men tend to accept significantly (*p* < 0.05) more stereotypes about black women, and to view women as “sexual objects.” Concerning “traditional gender roles” and media-related variables, no significant differences were observed (*p* > 0.05). Regarding these variables, a significant association (*p* < 0.05) was observed between watching “popular television” and violent movie programs. It was also verified that both variables would be good predictors (*p* < 0.001) of “making a move.” These results were confirmed in further analyses, using structural equation models (*p* = 0.13), which revealed an adequate fit. On the other hand, no gender differences were observed in these results (*p* > 0.05).

Another study, from Spain, conducted by Sánchez-Prada et al. [42] sought to assess attitudes and acceptance of IPV against women in 50 individuals. The results suggest that young adults tend to have less acceptance toward violence in three of the dimensions analyzed (i.e., the inferiority of women, victim-blaming, and minimizing and excusing abusers), except for “violence as a conflict resolution strategy”, where they showed more acceptance. It was also observed that gender had no impact on these results (*p* > 0.05).

Another study, also from Spain, conducted by Bonilla-Algovia and Rivas-Rivero [35] aimed at understanding the acceptance of maladaptive beliefs (MB) about gender roles (e.g., men are superior) and violence against women. The study was conducted with 2395 trainee teachers from Spain, Argentina, Chile, Colombia, El Salvador, and Mexico. The results suggest the existence of gender differences, with men presenting higher MB (*p* < 0.001) in all analyzed countries. Regarding differences between countries, it was observed that teachers from El Salvador and Colombia present more MB about gender roles and violence against women, than those of Mexico, Chile, Argentina, and Spain.

Finally, a study from Israel, conducted by Malka et al. [40], sought to analyze the beliefs of 542 students of Social Work, about physical violence against women. Most participants (90.5% to 98.7%) were strongly opposed to negative beliefs about physical violence against women/wives, namely: (1) there is a justification for the man to beat the woman; and (2) blaming the woman for the aggression. Being female, having more educated parents, more liberal and egalitarian attitudes toward women and the expectations associated with their marital role, and even witnessing less interpersonal psychological violence in childhood were shown to be associated with opposition to the first belief (*p* < 0.05). Being younger, having parents with a higher level of education, and less negative and illiberal attitudes toward women, as well as less expectations of unequal and patriarchal marital roles, were shown to be associated with opposition to the second belief (*p* < 0.001). Most participants also expressed a willingness to help assaulted women. Being more willing to do so seemed to be associated with factors such as: higher parental education level, higher socioeconomic family level, and less negative, illiberal, patriarchal, and inegalitarian attitudes towards women and the expectations of the marital role (*p* < 0.001). Similar results were found for beliefs that women were harmed by physical aggression and the willingness to hold violent husbands accountable for their behavior. These results were also shown to be associated with factors such as: being female (*p* < 0.001), having parents with a higher level of education (*p* < 0.05), and/or having more non-negative and liberal attitudes towards women and marital roles (*p* < 0.001).

### 3.2. Sexual Consent

Three studies from the USA addressed the evaluation of consent for sexual engagement. King et al. [39] sought to understand how non-verbal behaviors influence the perception of consent in 550 participants. Conversely to studies referring to other issues, in which the midpoint of the Likert scale is considered neutral, following the methodology of studies about consent through non-verbal signals [45], the lowest score (i.e., 0) was considered as “absolute non-consent”, with the remaining scores indicating subjectively stronger consent. Thus, significant differences were observed between genders (*p* < 0.001), with 43.1% of women answering “0 (i.e., definitely not), to all 26% proposed behavioral combinations, against only 20.3% of men. Men consistently obtained higher mean scores (*p* < 0.001) for all, behaviors or combinations thereof, indicators of consent.

Another study, carried out by Jozkowski et al. [15] with 185 participants, sought to understand gender differences in the definition of consent, as well as the way it is transmitted and understood. The results obtained suggest the absence of gender differences (*p* > 0.05) in the definition of consent. Yet, 16.2% of participants stated that consenting is “saying yes to sex.” Regarding the communication and interpretation of consent, the results suggest that participants showed a greater tendency to accept verbal consent indicators (*p* = 0.022), rather than non-verbal ones (*p* < 0.001). However, gender differences were observed (*p* < 0.05). Thus, men showed a greater acceptance of non-verbal indicators, while women showed a preference for verbal indicators of consent. Regarding non-consent, gender differences were also found (*p* < 0.05): men tended to emphasize non-verbal cues, while women showed a preference for a combination of verbal and non-verbal cues. Regarding specific sexual behaviors, participants revealed a preference for non-verbal indicators for less intimate behaviors (e.g., kissing, caressing). Conversely, for more intimate sexual behaviors, involving vaginal (*p* = 0.029), or anal penetration (*p* = 0.17), the preference was for verbal consent. Additionally, it was observed that men, when compared to women, resorted more to deceptive or aggressive tactics to obtain consent (*p* < 0.05).

Another study, conducted by Shafer et al. [28] aimed at understanding the attitudes, intentions, and interpretations of sexual consent of 301 men. The results indicated a significant negative association (*p* < 0.001) between attitudes of consent and beliefs in token resistance (i.e., boys’ false belief that, in the imminence of sexual contact, girls say no, when they would mean yes), and a positive association with the assertiveness of sexual communication (*p* < 0.001). No association was observed between hypermasculinity and the rape myth. Regarding intentions to obtain and respect consent, a negative association was observed with the rape myth (*p* < 0.01), and a positive association with assertive communication (*p* < 0.001). No association was observed with hypermasculinity nor symbolic resistance beliefs (*p* > 0.05). Regarding the interpretation of sexual aggression in consent scenarios, a negative association with token resistance (*p* < 0.001), and a positive association with the assertiveness of sexual communication (*p* < 0.001) were observed. No association with hypermasculinity or acceptance of the rape myth was observed. Regarding the interpretation of sexual consent in complex scenarios, negative associations with beliefs in token resistance (*p* < 0.001) and with the acceptance of the rape myth (*p* < 0.01), and positive associations with assertive communication (*p* < 0.001) were observed. Again, no association with hypermasculinity was found.

### 3.3. Exposure to and/or Watching Pornographic Content

Two studies sought to understand the relationship between watching pornographic content, on the Internet, or otherwise, the underlying beliefs and their behavioral consequences. The first, from Australia, was conducted by Bernstein et al. [34] with 385 participants and sought to understand the relation between watching pornographic content, and gender or sexual coercion beliefs and attitudes. The results suggest that most young people (41.3%) have early contact with pornographic content (i.e., at 12 and 14 years old), with men starting earlier, and that 81.3% continue seeking it out actively, with a higher incidence in men. It should also be noted that early initiation is positively and significantly related to the frequency of exposure (*p* < 0.001). Regarding the influence of pornographic content on sexual beliefs, no gender differences were observed (*p* > 0.05), despite slightly higher scores for women. Regarding the problematic watching of pornographic content, it was positively associated with gender stereotypes (*p* < 0.01), sexual impulsivity (*p* < 0.01), congruent beliefs with pornographic content (*p* < 0.01), early onset (*p* < 0.05), and watching frequency (*p* < 0.01). Regarding possible gender differences in the different analyzed variables, significant differences were only observed in sexual impulsivity (*p* = 0.006), with higher values for men. Finally, they tried to identify predictors of problematic watching of pornographic content. A model consisting of sexual impulsivity, congruent beliefs with pornographic content, and gender stereotyped beliefs was found, with a total explained variance of 28.7% (*p* < 0.001).

A second study, from Germany, performed by Krahé et al. [16] with 1181 participants, sought to understand the relation between watching pornography on the Internet and sexual risk behaviors. The results indicate that men tend to present higher levels of pornographic realism, risky assessments of consent to consensual sex, and sexual assault (*p* < 0.05). Women were more likely to suffer sexual assault, and of greater severity (*p* < 0.05). Regarding the impact of gender on risky sexual behavior or acceptance of sexual coercion, no differences were found (*p* > 0.05). Significant correlations were also observed, in both genders, between pornographic realism and risky assessments of consent to consensual sex (*p* < 0.001), acceptance of sexual coercion (*p* < 0.001), and risky sexual behaviors (*p* < 0.01).

### 3.4. Domestic Violence

Two studies related to domestic violence were included. The first, conducted in Turkey, by Adıbelli et al. [32], sought to understand the attitudes toward domestic violence (DV) of a group of men carrying out military service. The results suggested that 75.1% of the men did not have any knowledge related to DV, and that 41.2% thought that it was not necessary to report witnessed acts of domestic violence to the authorities. Additionally, 89.6% declared that they had never witnessed scenes of DV. Group comparisons suggested that there were no differences (*p* > 0.05) between some sociodemographic variables (e.g., age, occupation, type of family, marital status, offspring) and attitudes towards DV. Conversely, significant differences were observed (*p* < 0.05), on variables such as level of education, or having information about DV, with those who had lower levels of education tending to have more negative attitudes. The same comparison was made between those who declared that they would report any acts of DV to the authorities, and those who would not, with the older participants presenting more positive attitudes (*p* < 0.05).

A second study, from the USA, and carried out by Rodriguez et al. [26] with a group of 265 Indian men and women, aimed at evaluating the interactions between values, attitudes, and gender beliefs underlying domestic violence (DV). The results of this study, suggest the existence of gender differences between the belief systems related to partner aggression among young Indian adults, with a special emphasis on gender ideology, the importance of the male role in relationships, and the legal and social consequences of DV. In contrast, the moral values that influence women’s perceptions of DV are more complex and related to multiple beliefs about women’s power, family structure, and social and legal implications of DV. Additionally, they also suggest that women have a belief system with a more complex structure than men.

### 3.5. Conflict Resolution

A study, from Lithuania, performed by Aušraitė and Žardeckaitė-Matulaitienė [33] with 145 participants sought to understand the relationship between irrational beliefs (IB) and conflict resolution strategies. Four IB were considered: (i) “disagreement is destructive”; (ii) “partners cannot change”; (iii) “genders are different”; and (iv) “sexual perfectionism”. Preliminary analyses indicate that there is no association (*p* > 0.05) between IB and conflict resolution strategies and sociodemographic variables (e.g., age, gender, or duration of romantic relation). Further analyses revealed associations (*p* < 0.05) between IB and conflict resolution strategies. To illustrate, the most frequently cited IB, “disagreement is destructive”, was found to be positively and significantly associated with conflict resolution strategies “domain and avoidance”, and negatively associated with “integration and commitment”. “Partners cannot change” was positively associated with “avoidance” conflict resolution strategies, and negatively associated with “compromise” and “integration”. “Genders are different” was only positively associated with “dominance”. Finally, “sexual perfectionism” was negatively associated with “integration”.

### 3.6. Other Issues

A study from Australia, carried out by Towler et al. [43], sought to assess gender differences in the dynamics and severity of warning sign behaviors (WSB). Initial analyses only revealed associations (*p* < 0.05) between gender and WSB (i.e., dominance-possessiveness, denigration, and conflict-retaliation). Regarding gender differences in the perception of WSB severity, the average scores indicated that both genders perceived them as either serious or very serious. However, it was observed that women considered all WSB to be significantly more severe (*p* < 0.001). Additional variance analyses revealed that both genders viewed conflict retaliation as the most severe WSB (*p* = 0.017), followed by denigration and dominance-possessiveness.

A study conducted by Wagener and Padmanabhanunni [9] in South Africa, with a sample of 255 people, sought to understand gender differences in beliefs about aggression. According to the results, the beliefs that received the most approval from the participants were “verbal retaliation” and “retaliation against men.” Regarding gender differences in total scores, significant differences were observed (*p* < 0.05), with higher values for men. Regarding the scores, it was observed that men had higher values for the “approval of general aggression” belief (unprovoked) (*p* < 0.01). Regarding the remaining beliefs, no significant differences were observed (*p* > 0.05).

Another study from the USA, carried out by Canan et al. [36], sought to compare supportive attitudes towards sexual assault in Greek (i.e., involvement in fraternities or sororities) and non-Greek university students in a sample of 981 participants. The results suggest that overall, age (i.e., being junior or senior) (*p* = 0.025) and being Caucasian (*p* = 0.013) are associated with lower acceptance of token resistance (TR). Additional analyses, including variables of study (i.e., gender and Greek membership), suggest that age loses statistical significance (*p* = 0.097), as opposed to ethnicity (i.e., Caucasian) which gains significance (*p* < 0.001). Both variables, together, are associated with greater TR acceptance, particularly among men and those with a Greek affiliation (both *p* < 0.001). Despite minor changes in coefficients, it was found that the inclusion of the interaction term between gender and Greek membership into the model did not result in significant changes (*p* > 0.05). Gender moderated the association between Greek membership and TR acceptance, with Greek men increasing their acceptance (*p* = 0.029). Regarding the rape myth (RM) acceptance, the results suggest that age (*p* = 0.039), and being Caucasian (*p* < 0.001), are associated with lower RM acceptance. The inclusion of gender and Greek membership changes the association between first- and second-year students and the RM acceptance, which suggests that any differences between these two student groups and the rest can be explained by gender or by Greek nationality. However, gender and Greek membership is associated with increased RM acceptance, with Greek men revealing higher values. As seen in previous models, this one also reveals that gender moderates the relationship between Greek membership and RM acceptance, with Greek man showing greater RM acceptance (*p* = 0.009).

Finally, a study in Spain conducted by Durán and Rodríguez-Domínguez [38], sought to understand the impact of the sexist attitudes of college students on social reactions to two scenarios of sexual cyberviolence (i.e., at the end of their relationship, a man discloses sexual images of the ex-partner without her consent, either with or without implying that she was unfaithful). It was found that participants with higher levels of benevolent sexism (i.e., positive but stereotyped affective conception towards women, limited to the role of mother and/or wife), and hostile sexism (e.g., violent) blamed the victim more than those with lower levels. This result was particularly significant when the presented scenario mentioned that the victim had been unfaithful (*p* < 0.05). Regarding perceived severity, participants with higher levels of benevolent sexism assessed the situation as less severe, though this result was intensified in cases involving unfaithful victims/ex-partners (*p* < 0.001). Likewise, participants with higher levels of beliefs in benevolent sexism (*p* < 0.001), and hostile sexism (*p* < 0.05), tended to justify cyberviolence more than those with lower levels. The result was intensified when the victim was found to be unfaithful to an ex-partner (*p* < 0.001). In this sense, participants with higher levels of benevolent sexism and hostile sexism were also less likely to identify/report the situation presented as a case of gender-based violence (*p* < 0.05).

## 4. Discussion

The purpose of this SLR was to gain insight into the relationship between beliefs and violent behavior and to identify the most prevalent beliefs within its target population: young adults. This review identified several beliefs (e.g., rape myth, sexism, hypermasculinity) that serve to legitimize IPV, domestic and sexual violence (DV), and attitudes toward sexual consent, among others.

Concerning beliefs associated with IPV, Dardis et al. [37] suggest that women are more likely to suffer from sexual violence (SV), a finding that is in line with the results obtained by several authors (e.g., Chen et al. [46], who state that women are the main victims of SV). On the other hand, men are more likely to be victims of both physical and psychological violence, a result consistent with Lysova and Dim [47], who suggest that men are also victims of violence. Certain perspectives, notably those associated with hypermasculinity [48] suggest that men are entitled to exercise dominance over women. The results obtained appear to align with these beliefs, as violence against men is perceived as being more serious. The result by McKool et al. [48] suggest a similar direction, since most men reported having committed physical and sexual violence, some repeatedly, and at the same time they tended to downplay the severity of their actions. The fact that men tend to have more maladaptive gender beliefs may also contribute to this, as observed by Bonilla-Algovia and Rivas-Rivero [35]. Meanwhile, Malka et al. [40] observed a strong disapproval of more negative gender beliefs, associated acts of violence or victim blaming, which can be explained by the fact that the sample was composed mainly of women. Moss et al. [40], in turn, observed a strong acceptance of the objectification of women and traditional gender roles, which seemed to confirm the results obtained by several studies (e.g., D’Urso et al. [49] showing the prevalence of the belief in male superiority). The authors also noted the negative influence of some media content, which contradicts the results of recent studies (e.g., Abbas et al. [50]) confirming the existence of potentially negative effects from watching this type of content, which may contribute to the increase in violent acts). On the opposite side of the spectrum, the results of Sánchez-Prada et al. [42]) indicate that, compared to other age groups, young adults tend to be less accepting of violence, across all dimensions analyzed. Spain is an economically and socially advanced country. Therefore, the potentially positive effects of the respective culture should not be discarded [51]. However, as this is a small sample, these results should be interpreted with some caution.

Regarding consent to engage in sexual activities, the results observed by King et al. [39] indicated gender differences, with women responding negatively in greater numbers. The still widely prevalent belief in hypermasculinity that men should dominate relationships and reveal a greater predisposition for sex may help support these results [52]. Jozkowski et al. [15], in turn, did not observe a greater predisposition for men to accept non-verbal indicators, unlike women, who demonstrated a preference for verbal indicators. According to Chen et al. [46], victims of sexual violence are mainly women. This fact may lead them to adopt a more cautious stance when it comes to potential intimate involvement and to verbally communicate their consent. Sexual activities are increasingly common and natural in various contexts. However, this can lead to misunderstandings regarding the perception of consent, especially in cases involving the use of psychoactive substances. [53]. The potential for misunderstandings and resulting legal problems may help explain the need for assertive communication of consent, as observed by Shafer et al. [28]. Regarding beliefs associated with problematic viewing of pornographic content, Bernstein et al. [34] observed the growing and increasingly early contact of young people with this type of content, especially among males. Problematic viewing of such content has also been associated with gender stereotyping and sexual impulsivity. Szymkowiak et al. [54] suggested that young people, from an early age, have easy access to the Internet and new information technologies, which is in line with the results obtained by Krahé et al. [32], who also observed an association between viewing such content and risky sexual behavior, acceptance of sexual coercion and dangerous consent assessments. The increasing proliferation of sexual content on the Internet, much of which promotes aggressive sexual behavior or the subjugation of women [55], coupled with easy access [54], may explain these results.

Regarding domestic violence (DV), the results obtained by Adıbelli et al. [32] suggested little or no knowledge about the phenomenon. Rodriguez et al. (2021), on the other hand, observed the existence of a complex belief system that legitimizes DV. It is important to note that there is a consensus among authors that violence is more prevalent in social environments marked by poverty and degradation, as appears to be the case in India. Indian society is further characterized by a clear conservatism in the acceptance of a social structure that divides people into a “caste system”, and by the low socioeconomic and educational level of its population [56]. Together, these facts may explain these results.

Regarding other issues, Aušraitė and Žardekkaitė-Matulaitiene [53], indicated the negative impact of some irrational beliefs on conflict resolution strategies. These results are in line with Ellis [57] who suggested that irrational beliefs can restrict an individual’s information processing capacity. The ability to process available information, even if subtle, is essential (Araújo et al. [4]), as noted by Towler et al. [43], who suggest that women are more alert to subtle signs of potentially abusive behavior. As demonstrated by some authors (e.g., Sousa-Costa et al. [58]), women have historically, and continue to be, the main victims of violence, particularly the most serious ones (e.g., homicide). Therefore, it is possible to conjecture that, as Damásio [7] argues, they developed this ability as a result of their survival needs, or as a form of retaliation against possible aggressors, as Wagener and Padmanabhanunni [9] point out. Finally, as Davis-Williams [59] argued, the academic campus environment tends to favor acceptance and tolerance of inappropriate and non-consensual sexual behaviors (e.g., rape) against women. The results of Canan et al. [36], suggesting that male Greek-affiliated students were more inclined to accept the rape myth, appear to corroborate this idea. In short, the relationships of this population remain limited by maladaptive beliefs [4]. This was again confirmed by Durán and Rodríguez-Domínguez [38], who observed a positive association between sexism and victim-blaming (primarily directed at women), or the legitimization of cyber violence.

In summary, the results observed indicate the existence of a pattern that suggests the existence of significant associations between beliefs, namely the most undesirable ones, and undesirable behaviors (e.g., violence, antisocial behaviors). This is in line with the findings of several authors (e.g., Araújo et al. [4]; Rijo et al. [19]). Table 3 presents a summary of the main implications of this study.

### 4.1. Limitations and Potentialities

This study has some limitations, such as the fact that the samples are all composed primarily of college students, a population whose characteristics (e.g., social, cultural) may not align with those of the general population with identical demographic characteristics. Accordingly, while the results are undoubtedly relevant, they should be interpreted with some caution. It is also important to mention the small number of studies identified, which may be a limitation. However, it may also be a potentiality of this study, since as observed by Araújo et al. [4], this is still an unexplored field of study. Although beliefs have been the subject of considerable scientific interest, the majority of studies conducted have adopted a clinical perspective and have therefore focused on beliefs that underlie or contribute to the development of psychopathology. In addition, violence is not perpetrated solely by men. Thus, and according to the authors who suggest that certain beliefs (MB) promote violence, it would be relevant to carry out studies to better understand the most prevalent MB in each sex. Consequently, an understanding of the beliefs that support the most disruptive behaviors is of enormous importance and represents the most significant contribution of this study.

### 4.2. Implications for Future Investigations

Given the above, it would be important to carry out more studies on this topic, but with the recruitment of the general (i.e., non-academic) and forensic population, in order to understand the possible existence (or not) of differences between these groups.

### 4.3. Implications for Clinical Practice

The considerations made about the investigation can be extended to clinical practice, which tends to emphasize symptoms and observable behaviors. Most existing intervention models tend to follow the same path, with groups being divided up based on the same criteria. In this way, they may not respond to individual needs, especially in forensic contexts. This may explain the low effectiveness rates in reducing violence. Thus, the creation of intervention models based on the principles proposed in this study, with individuals divided into groups based on present beliefs, would be an asset, and could contribute to increasing the quality of the results obtained; that is, it could contribute to a drastic reduction in violence and the consequent number of victims. If it were to be achieved, it would open up a new and vast field of research and scientific production (Araújo et al. [60]).

## 5. Conclusions

This SLR once again highlights the negative impact of the most maladaptive beliefs on the violent behavior of young people. Tur-Prats [61] suggested that the magnitude and type of these effects depends on the culture analyzed. This fact becomes evident when observing the different manifestations of beliefs and violence, resulting from the scope of “latitudes” included in this study (e.g., USA, Turkey, Lithuania or Australia). So, some societies, especially those of some countries, other than those belonging to the so-called “Western countries”, tend to legitimize (e.g., India; Krishnakumar and Verma [62]), or even encourage (e.g., Saudi Arabia; Alquaiz et al. [63]) violence against women. It is also important to emphasize the relevance of people’s life contexts. According to Shishane et al. (2023), slums, generally frequented by people of low socioeconomic status, tend to be conducive to the outbreak of conflicts and violence, which may be aggravated in cases of psychoactive substance use.

Once again, it is also evident that men are also victims of violence, in their intimate or interpersonal relationships. In this way, they are often silent victims of the phenomenon who need urgent help. Thus, health professionals, psychologists included, must be particularly attentive. Men may ask for help following depressive or anxiogenic symptoms, the cause of which may be associated with victimization processes, in intimate or interpersonal relationships, but which, out of shame, they may not mention [64]. Another fact evidenced in this study, and in need of urgent attention by all those involved, is the growing and inappropriate use of new technologies, judging by the multiple reported cases of cyberviolence, in its multiple manifestations, such as cyberbullying, the dissemination of unauthorized content (e.g., intimate photos or videos), insults and threats or the increasingly early viewing of pornographic content). All this is facilitated by the premature access of children and young people to devices with Internet access (e.g., smartphones) [65]. In sum, this study contributes to alerting people to the need to combat the phenomenon of violence and, thus, increase the well-being of all.

## Figures and Tables

**Figure 1 ijerph-21-01500-f001:**
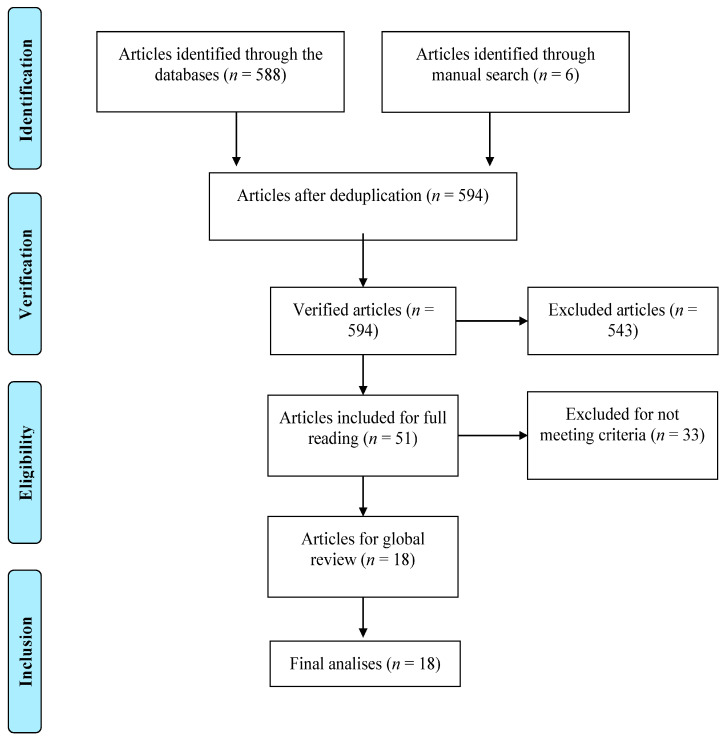
Flow Diagram.

**Table 1 ijerph-21-01500-t001:** Quality assessment of studies, according to the CCEERC Quantitative Research Assessment Tool.

	Population and Sample				Measurement		Analysis						
Study	Q1	Q2	Q3	Q4	Q5	Q6	Q7	Q8	Q9	Q10	Q11	Q12	Total
Adıbelli et al. (2019) [32]	1	1	0	1	1	1	1	0	1	1	1	0	9
Aušraitė, M., & Žardekkaitė-Matulaitiene (2019) [33]	1	1	−1	1	1	1	1	1	1	1	1	0	9
Bernstein et al. (2022) [34]	1	1	0	1	1	1	1	1	1	1	1	1	11
Bonilla-Algovia & Rivas-Rivero (2021) [35]	0	1	0	1	1	1	1	1	1	1	1	0	9
Canan et al. (2017) [36]	1	1	1	1	1	1	0	1	1	1	0	1	10
Dardis et al. (2017) [37]	1	1	1	1	1	1	1	1	1	1	1	1	12
Durán & Rodríguez-Domínguez (2019) [38]	0	1	0	1	1	1	1	1	1	1	1	0	9
Jozkowski et al. (2014) [15]	1	1	0	1	1	1	0	0	1	1	−1	1	7
King et al. (2020) [39]	1	1	1	1	1	1	1	1	1	1	0	1	11
Krahé et al. (2021) [16]	1	1	0	1	1	1	1	0	1	1	1	1	10
Malka et al. (2021) [40]	1	1	1	1	1	1	1	1	1	1	1	1	12
McKool et al. (2021) [41]	1	1	−1	1	1	1	0	1	1	1	NA	0	7
Moss et al. (2022) [17]	1	1	0	1	1	1	0	1	1	1	0	0	8
Rodriguez et al. (2021) [26]	1	1	0	1	1	1	1	1	1	1	1	0	10
Sánchez-Prada et al. (2020) [42]	1	1	0	0	1	1	1	1	1	0	1	1	9
Shafer et al. (2018) [28]	1	1	0	1	1	1	1	1	1	1	0	1	10
Towler et al. (2020) [43]	1	1	0	1	1	1	1	1	1	1	1	1	11
Wagener & Padmanabhanunni (2020) [9]	1	1	0	1	1	1	1	1	1	1	1	1	11

Note: NA: not applicable; Q1: Population; Q2: Randomized selection of participants; Q3: Sample size; Q4: Response and attrition rate: Q5: Main variables or concepts; Q6: Operationalization of concepts; Q7: Numeric tables; Q8: Missing data; Q9: Appropriateness of Statistical Techniques; Q10: Omitted variable bias; Q11: Analysis of main effect variables; Q12: Ethical approval.

**Table 2 ijerph-21-01500-t002:** Summary description of studies.

Study	Main Objective	Country of Origin	Participants (Age in Years)	Instruments	Main Results and Conclusions
Adıbelli et al. (2019) [32]	Assess attitudes toward DV of men performing military service.	Turkey	*N* = 221 men, *M_age_* = 23, *SD* = 3.62, range = 20–35.	ATDVS; QF;	- Negative attitudes towards DV affected by education level. - 10% of men exposed to DV.
Aušraitė, M., & Žardekkaitė-Matulaitiene (2019) [33]	Understand the relation between irrational beliefs and conflict resolution strategies.	Lithuania	*N* = 145, *n* = 110 women, *M_ag_*_e_ = 20.6, *SD* = nd, range = 18–29.	IRQB; ROCI;	- Irrational beliefs associated with increased use of inappropriate conflict resolution strategies.
Bernstein et al. (2022) [34]	Understand the relationship between viewing problematic IP, gender attitudes and sexual coercion.	Australia	*N* = 385, *n* = 270 women, *M_ag_*_e_ = nd, *SD* = nd, range = 17–25.	DEST; PHQ-9; SIC;	- Positive associations between IP-congruent beliefs, attitudes, gender stereotypes and impulsivity.
Bonilla-Algovia & Rivas-Rivero, (2021) [35]	Assess the acceptance of CD about gender roles and violence against women.	Spain	*N* = 2395, *n* = 1857 women, *M_age_* = 22.4, *SD* = 6.13, range nd.	IDTWUV-R; SQ;	- Significant differences between countries in the acceptance of CD. - Men with more CD regarding gender roles and violence against partners.
Canan et al. (2017) [36]	To assess attitudes of support to sexual assault in university students.	USA	*N* = 981, *n* = 630 women, *M_age_* = nd, *SD* = nd, range = 18–25.	IRMA-SF; SES-R *; SQ; TRSS;	- Gender, Greek membership, and race/ethnicity predictors of symbolic resistance and VM acceptance. - Superior effect in Greeks, and no gender differences in SV between groups.
Dardis et al. (2017) [37]	Understanding what behaviors are perceived as IPV.	USA	*N* = 703, *n* = 357 women, *M_ag_*_e_ = 18.9, *SD* = 1.06, range = 18–25.	CTS-2; MCSDS;	- Reports of IPV and abuse by men, and more serious reports by women - IPV perpetrated by men, considered the most abusive and the most severe, followed by SV and VP. - Men with a reduced perception of the severity of IPV.
Durán & Rodríguez-Domínguez, (2019) [38]	Assess the impact of sexist attitudes and transgression of the victim’s gender role on social perception of situations of sexual cyberviolence against women.	Spain	*N =* 301, *n* = 169 women, *M_age_* = 20.5, *SD* = 2.33, range = 18–27.	ASI; SARVA;	- Attitudes and transgression of gender roles attributed to women as predictors of more negative social evaluations regarding sexual cyberviolence. - Attitudes and transgression of gender roles, with a strong impact on the social perception of blaming victims.
Jozkowski et al. (2014) [15]	Understand how sexual consent is defined, and how it is communicated and understood.	UK	*N* = 185, *n* = 100 women, *M_age_* = nd, *SD* = nd, range = 18–31.	NSSHB; OEQ;	- No gender differences in the definition of consent, differing only in the content of the reports. - Women report more verbal strategies, and men tend to rely more on nonverbal strategies. - SV can arise from misinterpretation of consent.
King et al. (2020) [39]	Understand how non-verbal behaviors influence the perception of consent for sexual involvement.	USA	*N* = 550, *n* = 422 women, *M_age_* = 19.9, *SD* = nd, range nd.	QEPE; NVB;	- Men with more perceptions of consent through the interpretation of non-verbal behaviors.
Krahé et al. (2021) [16]	To assess the relationship between the frequency of IP use and the perpetration of sexual assault.	Germany	*N* = 1181, *n* = 762 women, *M_age_* = 25, *SD* = 3.52, range = 18–35.	ASC *; FUPPR; RSB *; RSS *; SAV.	- Realism of pornography predictor of involvement in risky sexual behavior and acceptance of coercion. - No gender differences in these results. - Indirect links between victimization and SV perpetration.
Malka et al. (2021) [40]	Understand the relationship between exposure to violence in childhood and the subsequent development of beliefs that legitimize violence against partners.	Israel	*N* = 542, *n* = 492 women, *M_age_* = 25.8, *SD* = 5.06, range = 20–57.	ATW; CTS-2; IBWB; MARI; SQ.	- No correlation between exposure to violence and negative beliefs about women. - Responsibility of the aggressors for the acts perpetrated against the partner. - Denial of responsibility to victims and expression of willingness to help them. - Significant correlation between gender egalitarian beliefs and intolerance of violence.
McKool et al. (2021) [41]	Assess the relationship between peer and self-reported perception of IPV perpetration.	USA	*N* = 202 men, *M_age_* = 22.6, *SD* = 3.4, range = 21–35.	ABM-SC; CTS-2; MPSIS; NIAAA; SQ.	- 36% and 67% of men reported having committed FV and SV in the last 12 months, respectively. - Most peers do not corroborate these reports. - Discordance variables associated with FV and VS reports.
Moss et al. (2022) [17]	Understand cognitions, social stereotypes and gender beliefs associated with IPV.	USA	*N* = 369, *n* = 185 women, *M_age_* = 28, *SD* = 6.27, range = 18–40.	AMDV; JSS *; SRSS; VQ *; WASO.	- Exposure to television and video, but not films, associated with acceptance of IPV. - Association moderated by traditional gender roles. - Sexual objectification of women and stereotypes about black women. - Exposure to “popular” films and series associated with IPV acceptance.
Rodriguez et al. (2021) [26]	Assess the interactions between values, attitudes and gender beliefs that support DV.	USA	*N* = 265, *n* = 142 women, *M_age_* = nd, *SD* = nd, range = 18–25.	ASBS; AWSA; IBAWB.	- Gender differences in beliefs related to DV. - Importance of gender ideology and the role of the male in relationships. - Women’s beliefs about their power in the family structure influence their DV perception. - Legal and social consequences of DV helps to understand beliefs and attitudes, in both genders.
Sánchez-Prada et al. (2020) [42]	Assess attitudes of acceptance of IPV against women.	Spain	*N* = 50, *n* = 25 women, *M_age_* = 19.8, *SD* = 2.30, range = 18–27.	IDTAWV; IBIPV; GVIAT; SQ.	- Young adults with less acceptance of attitudes of support for IPV against women. - This acceptance tends to increase with advancing age.
Shafer et al. (2018) [28]	Understand attitudes, intentions, and interpretations of sexual consent.	USA	*N* = 301 men, *M_age_* = 20.6, *SD* = 1.75.	SQD.	- Positive association between assertive communication and consent. - Hypermasculinity without any effect.
Towler et al. (2020) [43]	Understand gender differences in the perception of severity of warning signs for abusive behavior.	Austrália	*N* = 201, *n* = 152 women, *M_age_* = 21.1, *SD* = 2.59, range = 18–26.	HB; TREAD.	- Gender differences in perceptions of warning signs. - Women converge in the interpretation of these signs, as being worrying. - Both sexes perceive the partner’s retaliatory behaviors as being serious. - Women more aware of the warning signs, however subtle, for IPV as partner behaviors become more evident.
Wagener & Padmanabhanunni, (2020) [9]	Investigate the role of gender in (non) normative beliefs about aggression.	South Africa	*N* = 255, *n* = 199 women, *M_age_* = 20.2, *SD* = 1.44, range = 18–25.	NOBAGS; SQ.	- Men with more favorable beliefs about aggression in general, and with unprovoked aggression. - No gender differences regarding the higher approval of retaliation than unprovoked aggression.

Note. *: Adapted instruments; ABM-SC: Attachment to Abusive Male Peers subscale of Male Peer Support Questionnaire; AMDV: Attitudes Toward Male Physical Dating Violence Scale; ASC: Acceptance of Sexual Coercion; ASI: Ambivalent Sexism Inventory; ASBS: Adversarial Sexual Beliefs Scale; ATDVS: Attitude Toward Domestic Violence Scale; AWSA: Attitudes Toward Women Scale for Adolescents; CD: cognitive distortions; CTS-2: Revised Conflict Tactics Scale; DEST: The Dissociative Experiences Scale Taxon; DV: Domestic violence; FUPPR: Frequency of Use and Perception of Pornography as Realistic; FV: Physical violence GVIAT: Gender Violence Implicit Association Test; HB: Healthy and unhealthy behaviors; IBAWB: Inventory of Beliefs About Wife Beating; IBIPV: Inventory of Beliefs about Intimate Partner Violence; IDTAWV: Inventory of Distorted Thoughts about Women and Violence; IDTWUV-R: Inventory of Distorted Thoughts about Women and the Use of Violence-Revised; IRMA-SF: Illinois Rape Myth Acceptance Scale–Short Form; IPV: Intimate partner violence; IRBQ: Irrational Relationship Beliefs Questionnaire; JSS: Jezebel and Sapphire subscales; MCSDS: Marlowe-Crowne Social Desirability Scale; MPSIS: Male Peer Support’s Informational Support Subscale; nd: not disclosed; NIAAA: National Institute on Alcohol Abuse and Alcoholism’s; NOBAGS: Normative Beliefs About Aggression Scale; NSSHB: National Survey of Sexual Health and Behavior; NVB: Nonverbal Behaviors; OEQ: Open-ended Questions; PI: Problematic viewing of Internet pornography; PHQ-9:Patient Health Questionnaire; PV: Psychological violence QF: Questionnaire Form; SQD: Specially developed questionnaire; *SD*; standard deviation; SQ: Sociodemographic questionnaire; ROCI: Rahim Organizational Conflict Inventory–II; RSB: Risky Sexual Behavior; RSS: Risky Sexual Scripts; SARVA: Scale of Attribution of Responsibility to the Victim and the Aggressor; SAV: Sexual Aggression and Victimization Scale; SES-R: Sexual Experience Survey; SIC: Sexual Impulsivity Scale; SRSS: Sex Role Stereotyping Scale; SV: Sexual violence TREAD: Tendency to Resist or End Abusive Dynamics; TRSS: Token Resistance to Sex Scale; UK: United Kingdom; USA: United States of America VM: Violation myth; VQ: various questionnaires; WASO: Women Are Sexual Objects subscale from the Revised Attitudes Toward Dating and Sexual Relationships Measure.

**Table 3 ijerph-21-01500-t003:** Implications for Pratice, Research and Public Policies.

*Implications for Pratice, Research and Public Policies*
**For Practice** Early identification of maladaptive beliefs and understanding their relationship to violent behavior allows for rapid and effective intervention. Focus clinical practice on the beliefs underlying violent behaviors, along with observable behaviors. **For Research** Produce more and better research with other populations (e.g., ethnic minorities, LGBT, forensics). Seek to better understand the differences (e.g., sociodemographic and cultural characteristics, main beliefs) between these populations and university students, generally represented in most studies. Further exploration of gender differences in the perpetration of violence. **For Public Policies** Invest in primary prevention of violence, particularly in schools. Invest in the training of teachers and all educational assistants as frontline professionals with access to young people, to identify early signs of violent behavior and/or victimization. The fight against violence and the promotion of “normative” relations should be a priority in the education and civic formation of young people.

## Data Availability

As this is an SLR, data are only available in the original studies included.

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
