# Peer review of "Beliefs and Violent Behavior in Interpersonal Relationships of Young Adults: A Systematic Review"

_ijerph, 2024, doi:10.3390/ijerph21111500_

Round 1
Reviewer 1 Report
Comments and Suggestions for Authors
The article provides a comprehensive review of the role of beliefs in legitimizing violent behavior within young adult relationships. The research underscores how deeply ingrained cognitive schemas, often formed through maladaptive beliefs, can normalize or even justify interpersonal violence. This topic is highly relevant and expertly addresses pressing issues within psychology, behavioral sciences, and public health. The study’s systematic review approach, guided by PRISMA, adds rigor and credibility to the findings, as it carefully selects, evaluates, and synthesizes studies from diverse databases.
A key strength of the article is its strong methodological foundation. The structured process for identifying and selecting relevant studies ensures a high-quality dataset, enhancing the reliability of the results. The article contributes substantially to the literature by spotlighting beliefs that support intimate partner violence, domestic abuse, and sexual coercion, with an emphasis on rape myths and gender norms that perpetuate these behaviors. Furthermore, the article’s inclusion of geographically diverse data adds a valuable cross-cultural perspective, making its findings applicable to varied social contexts. By examining gender-based differences in beliefs, the authors provide insights that can inform targeted prevention strategies and policy frameworks aimed at mitigating youth violence.
However, while the article provides significant insights into the connections between beliefs and violent behavior, there are areas where the analysis could be expanded to address critical limitations. One area for improvement lies in the synthesis of beliefs, which largely covers broad belief categories but could further explore how these beliefs might interact with additional variables such as socioeconomic status, cultural background, or mental health, which are often substantial factors in violent behavior. Additionally, the article’s discussion on intervention strategies is limited, leaving a gap in practical applications. Considering the findings, a more in-depth examination of preventative or corrective approaches, particularly for maladaptive beliefs, would enhance the study’s practical relevance for practitioners and policymakers. Furthermore, the study’s exclusion criteria, which omitted longitudinal studies, could be reconsidered; these studies may offer insights into how violent beliefs and behaviors evolve over time, potentially providing critical data on the development and persistence of maladaptive beliefs that contribute to interpersonal violence. One notable aspect that appears lacking is a clear and well-defined conclusion. The article would benefit from a more explicit and structured summary that synthesizes the findings and implications of the review. Including a clear summary of the study’s main insights and actionable recommendations would enhance the overall cohesion and accessibility of the research, particularly for readers seeking to understand the practical takeaways. A well-defined conclusion would help bridge the detailed analysis with real-world applications, ensuring the study’s findings are not only informative but also readily applicable to policy and intervention planning.
My recommendation is to accept the article after minor revisions, with the above suggestions provided to the authors for consideration.
Author Response
Dear Reviewer,
Thank you for the speed of your review, as well as for your comments and suggestions, some of which are very pertinent. We try to respond to them in the best way, as far as we can, naturally constrained by the time available. In situations where we have not been able to accept their suggestions, we try to give an adequate justification or defend our views.
However, we believe that, thanks to your contributions, this manuscript has been substantially improved.
The Authors
Then we ask your questions (Q) accompanied by our answers (A)
Q1. The article provides a comprehensive review of the role of beliefs in legitimizing violent behavior within young adult relationships. The research underscores how deeply ingrained cognitive schemas, often formed through maladaptive beliefs, can normalize or even justify interpersonal violence. This topic is highly relevant and expertly addresses pressing issues within psychology, behavioral sciences, and public health. The study’s systematic review approach, guided by PRISMA, adds rigor and credibility to the findings, as it carefully selects, evaluates, and synthesizes studies from diverse databases.
Q2. A key strength of the article is its strong methodological foundation. The structured process for identifying and selecting relevant studies ensures a high-quality dataset, enhancing the reliability of the results. The article contributes substantially to the literature by spotlighting beliefs that support intimate partner violence, domestic abuse, and sexual coercion, with an emphasis on rape myths and gender norms that perpetuate these behaviors.
Q3. Furthermore, the article’s inclusion of geographically diverse data adds a valuable cross-cultural perspective, making its findings applicable to varied social contexts. By examining gender-based differences in beliefs, the authors provide insights that can inform targeted prevention strategies and policy frameworks aimed at mitigating youth violence.
A.1, 2, 3. We appreciate and are grateful for these comments. Our interpretation of them is that no changes will be necessary. Please correct if our interpretation is wrong.
Q4. However, while the article provides significant insights into the connections between beliefs and violent behavior, there are areas where the analysis could be expanded to address critical limitations. One area for improvement lies in the synthesis of beliefs, which largely covers broad belief categories but could further explore how these beliefs might interact with additional variables such as socioeconomic status, cultural background, or mental health, which are often substantial factors in violent behavior.
A4. We agree with you. We hope that the conclusion, already included, respond to your concerns.
Q5. Additionally, the article’s discussion on intervention strategies is limited, leaving a gap in practical applications. Considering the findings, a more in-depth examination of preventative or corrective approaches, particularly for maladaptive beliefs, would enhance the study’s practical relevance for practitioners and policymakers.
A6. We understand your worries. So, we made some changes (please see p.17. lines 603 to 606, 618, and 622 to 623).
Q6. Furthermore, the study’s exclusion criteria, which omitted longitudinal studies, could be reconsidered; these studies may offer insights into how violent beliefs and behaviors evolve over time, potentially providing critical data on the development and persistence of maladaptive beliefs that contribute to interpersonal violence.
A6. Thank you for your suggestion. However, the objective was to understand how the phenomenon manifests itself, and not its evolution over time. Furthermore, the authors conducted and published an RSL related to the same themes, to identify intervention programs, in which only longitudinal studies were included. For this reason, and because it seems redundant to us, longitudinal studies were excluded (i.e., exclusion criteria).
Q7. One notable aspect that appears lacking is a clear and well-defined conclusion. The article would benefit from a more explicit and structured summary that synthesizes the findings and implications of the review. Including a clear summary of the study’s main insights and actionable recommendations would enhance the overall cohesion and accessibility of the research, particularly for readers seeking to understand the practical takeaways.
Q8. A well-defined conclusion would help bridge the detailed analysis with real-world applications, ensuring the study’s findings are not only informative but also readily applicable to policy and intervention planning.
A7 and A8. Thanks for your suggestion. A conclusion is now included.
My recommendation is to accept the article after minor revisions, with the above suggestions provided to the authors for consideration.

Reviewer 2 Report
Comments and Suggestions for Authors
General Evaluation
This manuscript tackles a crucial and timely topic, particularly given the growing societal concern over violence in young adult relationships. It addresses an underexplored area by examining the role of beliefs in fostering violence, which is an important contribution. However, the manuscript lacks clarity in its objectives, suffers from structural inconsistencies, and could benefit from a more robust analysis of results and recent literature.
Strengths of the Manuscript
Relevance and Timeliness: The focus on violent behavior in young adult relationships is particularly pertinent, especially in the context of rising levels of such violence globally. The review’s focus on beliefs as a determinant of violent behaviors is critical for designing effective interventions. This is well-aligned with the findings from What do young people think about teen dating violence? A cross-cultural perspective, where the influence of cultural beliefs on violent behavior is highlighted​.
Contribution to the Field: By focusing on beliefs, the manuscript offers valuable insights into how cultural and personal norms may drive violent behavior in young adult relationships. This is in line with contemporary research exploring the intersection between societal norms and digital behavior, such as the findings presented in Impact of ICT and social networks on students​.
Areas for Improvement
1. Clarity and Precision of the Research Objective
Comment: The research question is not clearly defined. While the topic is important, the specific aim of the study is somewhat ambiguous.
Suggestion: The objective should be clearly and explicitly articulated. Consider formulating a precise research question that identifies the gap in the literature that this review aims to fill. Take inspiration from the methodological clarity in Face-to-Face Versus Online Harassment of European Women, which lays out its research focus in a succinct and structured manner​.
2. Structural Consistency
Comment: The manuscript’s structure is uneven, particularly in the methodological section, where the selection criteria for the studies reviewed are not clearly outlined.
Suggestion: Reorganize the methodology to include a detailed explanation of the study inclusion/exclusion criteria. Additionally, use more effective subheadings to enhance the flow of the review, as demonstrated in Strategies for Implementing GlobalConsent to Prevent Sexual Violence in University Men (SCALE)​.
3. Depth of Results Analysis
Comment: The discussion of results is superficial. The manuscript could benefit from a more thorough comparative analysis of different demographic factors (e.g., gender, culture) and their influence on violent behavior.
Suggestion: Consider using visual aids, such as charts or tables, to summarize the findings and highlight key differences between subgroups. This approach is effectively employed in Impact of ICT and social networks on students, where the use of gender-based analysis is clearly presented​.
4. Inclusion of Recent and Relevant Literature
Comment: The manuscript cites relevant studies, but some are outdated. More recent research on the role of beliefs and violence, especially in the context of digital behavior, should be incorporated.
Suggestion: Include more recent sources, particularly those that address the intersection of digital behavior and violence, such as Cyberbullying and Education: State of the Art and Bibliometric Analysis​. Incorporating these references would strengthen the manuscript’s relevance and scholarly depth.
5. Practical Implications
Comment: While the manuscript mentions potential practical applications, these are underdeveloped.
Suggestion: Expand the section on practical implications by offering more concrete suggestions on how the findings can inform policy and educational interventions. The comprehensive socio-educational approach outlined in What do young people think about teen dating violence? could serve as a useful model​.
6. Academic Writing and Style
Comment: The writing is generally clear but lacks sophistication in some sections. The language could benefit from greater precision and formal academic tone.
Suggestion: Revise the manuscript to improve clarity and precision, focusing on avoiding repetition and ensuring a more formal tone throughout. For example, Cyberbullying and Education provides a clear and well-structured narrative that can serve as a model for enhancing the writing quality of this review​.
References
González-Moreno, M. J., et al. (2020). Face-to-Face Versus Online Harassment of European Women: Importance of Date and Place of Birth. Sexuality & Culture, 24(1), 157–173. https://doi.org/10.1007/s12119-019-09632-4
Justification: This study explores how cultural and temporal factors influence harassment behaviors among young adults. Including this reference would enhance the manuscript's discussion on the role of beliefs and cultural contexts in shaping violent interpersonal relationships.
Yount, K. M., et al. (2024). Strategies for Implementing GlobalConsent to Prevent Sexual Violence in University Men (SCALE): Study Protocol for a National Implementation Trial. Trials, 25(1), 571. https://doi.org/10.1186/s13063-024-08401-5
Justification: This paper provides valuable insights into intervention strategies aimed at preventing sexual violence among university-aged men. Including this reference would strengthen the manuscript's discussion on effective prevention and educational programs targeting young adults.
Author Response
Dear Reviewer,
Thank you for the speed of your review, as well as for your comments and suggestions, some of which are very pertinent. We try to respond to them in the best way, as far as we can, naturally constrained by the time available. In situations where we have not been able to accept their suggestions, we try to give an adequate justification or defend our views.
However, we believe that, thanks to your contributions, this manuscript has been substantially improved.
The Authors
Then we ask your questions (Q) accompanied by our answers (A)
General Evaluation
This manuscript tackles a crucial and timely topic, particularly given the growing societal concern over violence in young adult relationships. It addresses an underexplored area by examining the role of beliefs in fostering violence, which is an important contribution. However, the manuscript lacks clarity in its objectives, suffers from structural inconsistencies, and could benefit from a more robust analysis of results and recent literature.
Strengths of the Manuscript
Relevance and Timeliness: The focus on violent behavior in young adult relationships is particularly pertinent, especially in the context of rising levels of such violence globally. The review’s focus on beliefs as a determinant of violent behaviors is critical for designing effective interventions. This is well-aligned with the findings from What do young people think about teen dating violence? A cross-cultural perspective, where the influence of cultural beliefs on violent behavior is highlighted​.
Contribution to the Field: By focusing on beliefs, the manuscript offers valuable insights into how cultural and personal norms may drive violent behavior in young adult relationships. This is in line with contemporary research exploring the intersection between societal norms and digital behavior, such as the findings presented in Impact of ICT and social networks on students​.
Note. We don´t have considered dating violence, because that is a typical from adolescent people.
Areas for Improvement
Q1. Clarity and Precision of the Research Objective
Comment: The research question is not clearly defined. While the topic is important, the specific aim of the study is somewhat ambiguous.
Suggestion: The objective should be clearly and explicitly articulated. Consider formulating a precise research question that identifies the gap in the literature that this review aims to fill. Take inspiration from the methodological clarity in Face-to-Face Versus Online Harassment of European Women, which lays out its research focus in a succinct and structured manner​.
A1. Thanks for your advice. We decide to change the formulation of the Research questions. In our opinion, there are clearer now. Please see p 4, lines 144 60 154.
Q2. Structural Consistency
Comment: The manuscript’s structure is uneven, particularly in the methodological section, where the selection criteria for the studies reviewed are not clearly outlined.
Suggestion: Reorganize the methodology to include a detailed explanation of the study inclusion/exclusion criteria. Additionally, use more effective subheadings to enhance the flow of the review, as demonstrated in Strategies for Implementing Global Consent to Prevent Sexual Violence in University Men (SCALE)​.
A2. We understand your concerns, but we don’t agree. In our opinion this section complies all the guidelines PRISMA, as suggested by reviewer 1. So, we decided to not make changes, because we have no time, and one author is not available (she is ill). For this, we present our apologizes.
Q3. Depth of Results Analysis
Comment: The discussion of results is superficial. The manuscript could benefit from a more thorough comparative analysis of different demographic factors (e.g., gender, culture) and their influence on violent behavior.
Suggestion: Consider using visual aids, such as charts or tables, to summarize the findings and highlight key differences between subgroups. This approach is effectively employed in Impact of ICT and social networks on students, where the use of gender-based analysis is clearly presented​.
A3. Once again, we are very sorry, for the same reasons of A2.
Q4. Inclusion of Recent and Relevant Literature
Comment: The manuscript cites relevant studies, but some are outdated. More recent research on the role of beliefs and violence, especially in the context of digital behavior, should be incorporated.
Suggestion: Include more recent sources, particularly those that address the intersection of digital behavior and violence, such as Cyberbullying and Education: State of the Art and Bibliometric Analysis​. Incorporating these references would strengthen the manuscript’s relevance and scholarly depth.
A4. Once again, thanks for your suggestion. So, we decided to eliminate some older citations, and include new ones that, he hopes, address your concerns. Please see references section. New references are highlighted in green.
Q5. Practical Implications
Comment: While the manuscript mentions potential practical applications, these are underdeveloped.
Suggestion: Expand the section on practical implications by offering more concrete suggestions on how the findings can inform policy and educational interventions. The comprehensive socio-educational approach outlined in What do young people think about teen dating violence? could serve as a useful model​.
A5. You’re right. So, we made some changes as you suggested. Please see p, 17, lines 603 to 607, 618, 622 to 623 and the new subsection “Conclusion”, pp 17-18, lines 626 to 651.
Q6. Academic Writing and Style
Comment: The writing is generally clear but lacks sophistication in some sections. The language could benefit from greater precision and formal academic tone.
Suggestion: Revise the manuscript to improve clarity and precision, focusing on avoiding repetition and ensuring a more formal tone throughout.
A6. Once again, you’re right. We hope that you understand the fact that we are not native speakers of English. In addition, our language has a more complexes grammatical rules, that difficult the translation process. So, we decided to make a total review of the manuscript.

Reviewer 3 Report
Comments and Suggestions for Authors
The article presents a systematic review exploring the link between beliefs and violent behaviors in young adults' interpersonal relationships. The authors aim to identify prevalent beliefs associated with legitimizing violent behavior in intimate relationships. Although the paper is well structured and focused, some relevant issues and considerations are outlined below:
1. Although the authors indicate this in the introduction, the controversy over the range of subjects that can be included in the term "young adults" is not resolved. The authors present different points of view from different authors, widening the age range, but do not take a position on any of the options. Nevertheless, they state in the article that the study population includes subjects between 18 and 35 years of age. This choice must be justified, using one of the previous definitions in any case, and the reasons for this choice must be stated.
2. Although the overall aim of the review is clear, the focus could be better defined. Ideally, the use of frameworks such as PICO (population, intervention, comparison, outcome) will provide clarity and help to formulate precise questions.
3. A major limitation affecting the quality and transparency of the review is the lack of prior registration of the review in PROSPERO. The prior specification of both the objectives and the research question, as well as the criteria and methods of analysis, is essential, as any subsequent deviations should be properly justified. In this case, we do not have this information because it was not previously registered.
4. The flowchart figure should be improved for readability. I recommend providing the results of the studies by database, as well as the criteria used in each phase of elimination, along with the number of articles eliminated per criterion.
5. The article mentions the search in three databases (EBSCO, PubMed and Web of Science) and the evaluation of 594 studies, but there is insufficient detail on the specific search strategies used. A quality systematic review should not only detail the keywords and Boolean operators, but also the filters used. I also recommend recording the steps of the search strategy in a table, indicating the search number together with the keywords + Booleans + filters (if applicable) and the results up to the last search. This table can be included as supplementary material. This is, of course, a recommendation, since if the authors have not performed this procedure at the time of the review, this information cannot be obtained unless the review is performed again.
6. Although the CCEERC Quantitative Research Assessment Tool is used to assess methodological quality, no formal analysis of the risk of bias in the included studies was performed, which is a crucial step in a rigorous systematic review. Tools such as ROBINS-I can be used for this purpose.
Author Response
Dear Reviewer,
Thank you for the speed of your review, as well as for your comments and suggestions, some of which are very pertinent. We try to respond to them in the best way, as far as we can, naturally constrained by the time available. In situations where we have not been able to accept their suggestions, we try to give an adequate justification or defend our views.
However, we believe that, thanks to your contributions, this manuscript has been substantially improved.
The Authors
Then we ask your questions (Q) accompanied by our answers (A)
The article presents a systematic review exploring the link between beliefs and violent behaviors in young adults' interpersonal relationships. The authors aim to identify prevalent beliefs associated with legitimizing violent behavior in intimate relationships. Although the paper is well structured and focused, some relevant issues and considerations are outlined below:
Q1. Although the authors indicate this in the introduction, the controversy over the range of subjects that can be included in the term "young adults" is not resolved. The authors present different points of view from different authors, widening the age range, but do not take a position on any of the options. Nevertheless, they state in the article that the study population includes subjects between 18 and 35 years of age. This choice must be justified, using one of the previous definitions in any case, and the reasons for this choice must be stated.
A.1 This SLR is part of a set of three prepared by the authors, and more three empirical studies, which constitute the Doctoral Thesis of the first author. Given the aforementioned ambiguity regarding the limits of this age group, the authors, following the methodological options taken in the aforementioned studies, opted for these ages (i.e., 18-35). This decision was also based on the fact that, as far as we know, they are those used in literature. Thus, we chose to include an additional citation to justify the decision (Please see p.4, lines 153-154).
Q2. Although the overall aim of the review is clear, the focus could be better defined. Ideally, the use of frameworks such as PICO (population, intervention, comparison, outcome) will provide clarity and help to formulate precise questions.
A2. We agree with you on this topic. Thus, we chose to reformulate the research questions. (please see p.4, lines 150-152).
Q3. A major limitation affecting the quality and transparency of the review is the lack of prior registration of the review in PROSPERO. The prior specification of both the objectives and the research question, as well as the criteria and methods of analysis, is essential, as any subsequent deviations should be properly justified. In this case, we do not have this information because it was not previously registered.
A3. We understand your concern. However, it’s important to remember that this SLR, as well as another already published in a prestigious journal (cited in this SLR), began to be prepared in 2021/2022, a period that coincided with the COVID-19 pandemic. At the time, PROSPERO gave preference to the registration of studies related to COVID-19. So, other types of studies were systematically left behind. We hope that you will understand that we had deadlines to meet, so we could not wait. We assume that this is a limitation, but that it does not invalidate this work, just as it did not invalidate the one that is already published.
Q4. The flowchart figure should be improved for readability. I recommend providing the results of the studies by database, as well as the criteria used in each phase of elimination, along with the number of articles eliminated per criterion.
A4. Once again, we understand your concern and thank you for the warning that will be considered in the studies that are underway. As already mentioned, this work is more than two years old, currently, we have not been able to, or do not have the information you suggest adding. Allow me to also argue that, even though it is relevant, it is not mandatory to put the information you mention, as it will be possible to verify in multiple SLRs published in important journals. Nevertheless, we take responsibility for the error and apologize, hoping that it will not make the publication of this study impossible.
Q5. The article mentions the search in three databases (EBSCO, PubMed and Web of Science) and the evaluation of 594 studies, but there is insufficient detail on the specific search strategies used. A quality systematic review should not only detail the keywords and Boolean operators, but also the filters used. I also recommend recording the steps of the search strategy in a table, indicating the search number together with the keywords + Booleans + filters (if applicable) and the results up to the last search. This table can be included as supplementary material. This is, of course, a recommendation, since if the authors have not performed this procedure at the time of the review, this information cannot be obtained unless the review is performed again.
A5. Thank you for your comment, which seems pertinent to us. Thus, we inform that no type of filters was used, because the number of articles identified was not very high, something that facilitated the work of the authors. To clarify, we have chosen to mention this fact in the text (please see p. 5, lines 182-183).
Q6. Although the CCEERC Quantitative Research Assessment Tool is used to assess methodological quality, no formal analysis of the risk of bias in the included studies was performed, which is a crucial step in a rigorous systematic review. Tools such as ROBINS-I can be used for this purpose.
A6. Thank you for your suggestion. Research teams at our university generally use this tool, CCEERC. We agree that the possibility of biases is not properly taken care of. However, we trust that this will have been safeguarded in the review process of the articles included in this SLR. We also hope that you understand that the use of the tool you suggest could force you to redo the entire study, something that is not at all feasible at this time, due to the overload of work and the short time of this review process (i.e., 10 days).

Round 2
Reviewer 3 Report
Comments and Suggestions for Authors
Comments have been answered to the best of the authors' ability.